# An Abrupt Transition to Digital Teaching—Norwegian Medical Students and Their Experiences of Learning Output during the Initial Phase of the COVID-19 Lockdown

**DOI:** 10.3390/healthcare10010170

**Published:** 2022-01-17

**Authors:** Henriette K. Helland, Thorkild Tylleskär, Monika Kvernenes, Håkon Reikvam

**Affiliations:** 1Faculty of Medicine, University of Bergen, 5020 Bergen, Norway; Henriette.K.Helland@student.uib.no (H.K.H.); Monika.Kvernenes@uib.no (M.K.); 2Centre for International Health, University of Bergen, 5020 Bergen, Norway; Thorkild.Tylleskar@uib.no; 3Department of Clinical Science, University of Bergen, 5020 Bergen, Norway; 4Clinic for Medicine, Haukeland University Hospital, 5021 Bergen, Norway

**Keywords:** medical education, digital education, COVID-19 pandemic

## Abstract

Norwegian universities closed almost all on-campus activities on the 12 March 2020 following a lockdown decision of the Norwegian government in response to the COVID-19 pandemic. Online and digital teaching became the primary method of teaching. The goal of this study was to investigate how the transition to digital education impacted on medical students enrolled at the University of Bergen (UiB). Key points were motivation, experience of learning outcomes, and fear of missing out on important learning. Using an online questionnaire, students were asked to evaluate the quality of both lectures and taught clinical skills and to elaborate on their experience of learning output, examination, and digital teaching. Answers from 230 students were included in the study. Opinions on the quality and quantity of lectures offered and their experience of learning output varied based on gender, seniority and the amount of time spent on part time jobs. Students at UiB were generally unhappy with the quality of teaching, especially lessons on clinical skills, although both positive and negative experiences were reported. Securing a satisfying offer of clinical teaching will be important to ensure and increase the student experience of learning output in the time ahead.

## 1. Introduction

Norway introduced a general lockdown on the 12 March 2020, in response to the global COVID-19 pandemic [1]. Despite the comprehensive restrictions, the universities were encouraged to keep up the pace in the educational programs. Emergency remote learning (ERL), a temporary shift from the traditional form of education into a remote one following a state of emergency, was implemented by the Norwegian universities to ensure the continuation of higher education [2]. This abrupt transition to emergency remote learning was a worldwide phenomenon. More than 1.9 billion students from 190 countries were forced to transfer their education from face-to-face to digital education to fight the ongoing pandemic, according to UNESCO [3]. In Norway, emergency digital education included the transition from a mainly physical learning environment to video recordings, live lectures on digital platforms and home exams [4]. Prohibition of physical attendance left students without an office, without the possibility of hands-on learning of clinical skills, without academic and social meeting places and with major changes in their daily study habits [4]. Transitioning to emergency digital education has been challenging in most countries. A study conducted on middle school students in Palestine reported that the quality of emergency remote learning has been low even compared to digital learning in normal circumstances. Course designs, assessment, and teaching strategies in schools and at universities are originally designed for face-to-face teaching. In addition, both educators and students are living under a high level of stress, anxiety, and uncertainty due to the state of emergency affecting all parts of society and normal life [5]. The challenges of emergency digital education have potentially been similar for medical students in Norway. The abrupt closing of Norwegian universities meant that medical schools had little or no time to restructure their education and prepare for digital education. In addition, medical training facilities in Norway were closed, and students were left without a place to practice clinical skills.

Face-to-face education is today a crucial part of medical education worldwide [6]. Acquiring practical skills is of utmost importance when learning how to practice medicine. However, according to the World Health Organization (WHO) digital education may be capable of supplying the estimated 4.3 million shortage in healthcare workers worldwide [7]. Student experience of digital education during the pandemic may provide useful information on how to develop and implement a potentially more digitalized medical education in the future.

We conducted a study to investigate how the period from the 12 March 2020 to the end of the spring semester (20 June 2020) affected the lives and the learning environment for medical students at the University of Bergen (UiB) in Bergen, Norway. The aims were to assess students’ own experiences of learning output, motivation, and possible fear of missing out on important learning, due to the switch to digital education. In addition, we wanted the student’s perspective on the positives and negatives of digital education, and their opinion on how this type of education could be improved in the future. This study is the first study to explore the student experience of the COVID-19 pandemic on medical students in Norway.

## 2. Subjects and Methods

### 2.1. Undergraduate Medical Program at UiB

The undergraduate medical program at UiB is a six-year long program divided into 12 terms, referred to as MED1, MED2 etc. It is organized as three columns that run parallel throughout the program (columns of profession, academics, and professionalism) and include elements of a spiral curriculum where key topics and subfields are revisited progressively. Teaching is offered using primarily lectures, Team-based learning [8], and clinical teaching. There is a final assessment after each term using multiple choice (MCQ) and short answer questions (SAQ). Objective structured clinical examinations are held after the 3rd and 6th year.

### 2.2. Setting and Application

Our study is a cross-sectional retrospective study, using a combination of qualitative and quantitative research methodology. We used a questionnaire to collect data for assessing the educational environment. The questionnaire was administered to all undergraduate medical students at UiB between the 12th of June 2020 and the 16th of August 2020 using the digital tool Skjemaker^®^, a local adaptation of MachForm (Appnitro software, Malang, East Java, Indonesia, Available online: www.machform.com (accessed on 5 January 2022)). Students were invited to participate in our study through a link sent to their student e-mail portal with a copy sent to their private e-mail address. Data was collected in relation to the 2020 spring semester when restrictions were most intense. The questionnaire was first shared a few days after the last exam of the semester to ensure that students would be able to answer questions about the final exam with the least amount of memory bias. Three reminders were sent out during the data-collecting period in effort to boost the response rate. In addition, a link to the study was published in relevant social media. The questionnaire was completed in approximately 15 min. The study was approved by the Norwegian Center for Research Data (NSD) and used voluntary consent. Participants were informed that they had the right to discontinue the study at any point without any consequences. Answers to the questionnaire are completely anonymous, and investigators has no way of identifying participating students.

### 2.3. Sample Size Determination

All medical students at UiB (958 in total) were invited to participate, which means there was no sampling. This was also done to ensure inclusion of the variations in teaching methods and adaptation to digital education between all six years of the program.

### 2.4. Questionnaire

A self-designed questionnaire was used in this study. Design and layout of the questionnaire was discussed several times between authors to maximize face validate. A small pre-test group of five students were used to standardize questions. The participants offered some amendments to the questionnaire which were considered and noted. The questionnaire was finalized after an in-depth discussion among the authors. The end-product consisted of 53 questions divided into five different categories, (Table 1), each addressing different aspects of digital education. 27 questions used a Likert type response scale, 10 were closed-ended questions, and 16 questions were open ended. Quantitative information was collected using either a 5-point Likert scale [9] or close-ended questions. Open-ended questions were used to collect qualitative data explaining the rationale behind answers to the other questions. The questionnaire was in Norwegian. Quotes, tables, and diagrams reported in the following are translated from Norwegian to English for the purpose of this article.

### 2.5. Statistical Analysis

Quantitative data were analyzed using Stata (StataCorp, College Station, TX, USA). Comparison of ordinary data between groups was done using the Mann–Whitney U-test and the using Kruskal–Wallis test for analyzing more than two groups. Participants were categorized based on gender, years of study and whether they were providing care for children. Graded Likert scale questions are presented descriptively including mean values and a 95% confidence interval. Potential gender differences in student experience of digital education were investigated, as well as differences between years of studying medicine. Differences between students spending more, equal, or less hours at a part time job outside of studies was also investigated. Qualitative data were reviewed, sorted, and categorized thematically.

## 3. Results

### 3.1. Quantitative Response Data

A total of 230 students, a response rate of 24%, submitted answers to the questionnaire—169 women (73%) and 61 men (27%). The gender distribution among participants reflects the current gender distribution at the medicine program at UiB—75.7% women and 24.3% men [10]. Participation was somewhat uneven among students at different years of study. For instance, the response rate of students in their last semester (6th year, MED12) was 49% while the response rate of students in their second last semester (6th year, MED11) was only 13% (Table 2). 17 students cared for children during the lockdown, of which 13 were women and three were men. A total of 19% of students had spent more time, 41% less time and 40% equal amount of time in their paid part-time positions outside of study as before the lockdown. A total of 74 students were living with other students during the initial lockdown, and 102 students moved back in at their parents’ house for some period.

### 3.2. Quality of Digital Education Media

Different digital teaching methods, specifically pre-recorded PowerPoint slides with sound, pre-recorded video lectures, and live video lectures, were rated based on technical, academic, and pedagogical quality. Live video lectures had the highest mean score in pedagogical quality (3.69 (3.57–3.82)), while pre-recorded lectures scored the highest on both academic (4.02 (3.9–4.12)) and technical quality (3.94 (3.18–4.07)) (Figure 1). Female students were statistically significantly more satisfied with pedagogical quality of all digital education media than their male colleagues (Live video: *p* = 0.002. Video recording: *p* = 0.05. PowerPoint with sound: *p* = 0.03).

### 3.3. Teaching of Clinical Skills and Hands-on Education

The students were asked whether they agreed, disagreed or were indifferent to several statements regarding the quality of the clinical training. One statement read “In my experience, my benefit from clinical and practical education has been good compared to an ordinary, physical semester”, on which 61.7% of students disagreed, 23.0% were indifferent and 14.8% agreed. Another statement “I believe that our clinical and practical education has been replaced in a satisfying manner”, on which 59.1% of students disagreed, 22.6% were indifferent and 18.3% agreed (Figure 2). However, there was statistically significant differences between students in the different semesters. There was a statistically significant difference in both experience of benefit of clinical and practical lectures (*p* = 0.003), and in experience of satisfying replacement of clinical education (*p* = 0.003) between the different semesters (Figure 3).

### 3.4. Student Experience of Own Learning Output

Mean value of students’ perceived learning output was 2.59 (Figure 4). Experienced learning output compared to an ordinary, physical semester correlated positively with satisfying information about changes in timetable and lectures (*p* < 0.001), student experience of their own motivation for learning (*p* < 0.001) and student experience of their own study efforts (*p* < 0.001). Motivated students, with enough and satisfying information given during the semester and with high own study efforts, experienced higher learning output than their counterparts. In addition, students who put in more hours in their part time job than during an ordinary semester had a lower learning outcome than students with fewer or the same hours (*p* = 0.0025) and had a more negative attitude towards digital education (*p* < 0.001).

Female students had a more positive attitude towards digital education than their male colleagues (*p* = 0.04), while at the same time being more anxious of having lost out on important learning because of digital education (*p* = 0.05). This fear of losing out on important learning also differed between the semesters (*p* < 0.001). Semesters MED7 and MED10 were generally more worried about potential gaps in their knowledge, while MED9 and MED12 where less anxious.

### 3.5. Student Feedback on How to Improve Digital Education

Several pros and cons were highlighted when students were asked about their experience with digital education. A majority emphasized the possibility of structuring their own study days as the biggest advantage, including choosing when to watch a lecture, being able to repeat when needed and being able to pause or rewind in case of ambiguity. Reduced possibility of socialization, lack of structure and canceled classes were, on the other hand, emphasized as factors that students disliked with digital education (Table 3).

Several proposals were made on how to improve digital education in the future. Improving teachers’ technical skills, increased use of interactive teaching and improving the structure and flow of information on the learning management system (Canvas) (Available online: https://www.instructure.com/en-gb/canvas (accessed on 29 December 2021)), were some of the proposals made by several students (Table 4). In addition, students demanded that the faculty of medicine should ensure that all physical lectures receive some form of digital replacement, as opposed to just being cancelled. Digital education should correspond with the original timetable, and teachers should agree on one platform to present information to their students.

## 4. Discussion

Medical students at UiB reported an experience of lower learning output because of emergency digital education the spring semester of 2020. This is despite that increased use of digital education is generally found to be a positive contribution to the learning environment at a university. Most of medical students in our study did report several positive factors regarding digital education. These included the increased opportunity to structure their own days, increased flexibility, and experience of increased time efficiency. Learning values, such as flexibility, usefulness and worthiness are found to have a positive effect on behavioral intention in context of digital learning. Behavioral intention is defined as the motivational factors that influence a given behavior [11]. Greater behavioral intention increases the likelihood of a desired behavior. In this context, increased behavior intention with increased learning values means that students that experience learning value are more likely to utilize digital education and learning platforms in a productive way [12]. Other studies have reported that an increased sense of time efficiency among medical students related to digital education have led to increased family time, better quality of sleep and increased opportunities for research [13,14]. Video lectures have shown the possibility of increasing learning output and speed of learning, especially when watching speed can be regulated and when students may repeat a lecture if they like [15].

Medical students in our study did also report a general satisfaction with the quality of theoretical teaching provided. Instructor characteristics are defined as qualities that makes a good instructor or teacher [16]. Qualities such as being able to fully utilize an eLearning platform and answering questions, are good instructor characteristics, and are found to increase behavioral intention in students in context of digital learning [12]. However, medical students in our study did request a higher degree of familiarization of technical aids from their teachers. They also requested the teachers to ensure the possibility to ask questions. Absence of good instructor characteristics may have reduced behavioral intention in medical students at UiB and contributed to the experience of a low learning output. A study by Bettinger et al. also show that to some extent, online learning might not compete with aspects of other learning, such as interactive knowledge building between teacher and student [17]. This finding is consistent with our study, with 22.9% of students reporting that teachers should take better advantage of the possibility of interactive learning in digital education.

Our study showed a correlation between lack of motivation and reduced learning output. Previous studies have shown that learning motivation is directing towards achievement and is therefore an essential part of perceived learning output and academic success [18,19,20]. Lack of motivation may have several possible explanations. Reduced motivation in students during the COVID-19 pandemic has been linked to for instance the individual student surroundings. Students possessing good internet access, a quiet and suitable place to study and with a high degree of digital social interaction have a higher degree of motivation during the pandemic [21]. A study by Khlaif et al. shows that poor internet connection and lack of technical support affect digital education in a major way [5]. Most of students in our study were living with other students during the initial lockdown, and 102 students moved back to their parents’ house for some time. Lack of information from the university, moving back home to parents and constant changes in workload are likely linked to increased stress and consequently reduced motivation in for instance psychology students [22]. Living in dormitories with other students may also have caused a lack of a suitable study environment for students in our study. In addition, the unpredictability of emergency digital education and the pandemic in general may have caused an increased level of stress among students [5].

There was a major dissatisfaction with how clinical and practical teaching was replaced among medical students at UiB. Dissatisfaction with the practical teaching offered has been reported in several other studies, and several raise concern that the COVID-19 pandemic has resulted in a serious lack of clinical skills among medical students [14,23]. A study by AlQhani et al. showed that more than half of respondents thought that online learning was much or somewhat less effective in balancing practical and theoretical experience. Satisfaction was decreasing with increasing years of study and was especially low when practical aspects of teaching were at its highest [24]. A concern for the UiB students has been the forced cancellation of Objective Structured Clinical Examination (OSCE) for students in their final year of study. OSCE often serves as an additional motivation for students to practice clinical skills, and several studies fear that the cancellation of this exam will result in a generation of doctors unsure in their own clinical skills [25,26,27,28]. In this context, it should be mentioned that the graduating class in our study had generally low anxiety for having lost important skills and knowledge.

Our study showed a difference between male and female students. Female students were generally more satisfied with the pedagogical quality of teaching and had a more positive attitude towards more digital education in the future. At the same time, they were more anxious about potential important gaps in their knowledge. A study by Worly et al. has shown that female medical students have a higher risk of burnout and emotional exhaustion compared to male students [29]. In addition, female medical students have had a greater increase in stress levels and anxiety during the COVID-19 pandemic than their male colleagues [23]. A paradoxically more positive attitude towards digital education may be surfacing due to female students generally being more structured [30], and digital education demanding a higher degree of self-organization. However, earlier studies have not shown a difference in the usage of digital tools between male and female students [31].

A limitation to our study is that it did not assess students’ performance. Their perceived learning output may not be the same as their actual learning output. However, subjective feedback is essential to map out student opinions and to investigate how to make digital education a better experience. It is essential to develop digital education further. Our study had a low response rate, with 24% of medical students at UiB. At the same time, our response rate and final number of participants is consistent with similar published studies. Gender distribution is consistent with gender distribution on the medical education of UiB.

## 5. Conclusions

Despite several studies showing the positive potential of digital education in the field of medicine, medical students at UiB generally reported an experience of reduced learning output during the COVID-19 pandemic. Our findings indicate that lack of motivation and lack of a sufficient offer in clinical education are the biggest contributing factors. Practical and clinical skills are essential to the field of medicine, and a lack of opportunity to rehearse and practice skills could potentially lead to a generation of insecure doctors with reduced experience in meeting and examining actual patients. However, if used correctly, digital education can be a most useful tool to increase flexibility and time efficiency among students and could even contribute to an increased learning output. With an increased focus on securing student motivation, learning values, and good instructor characteristics, as well as investigating and utilizing tools for clinical digital education, digital education may prove to be a most useful tool for educating medical students and other health workers in the future.

## Figures and Tables

**Figure 1 healthcare-10-00170-f001:**
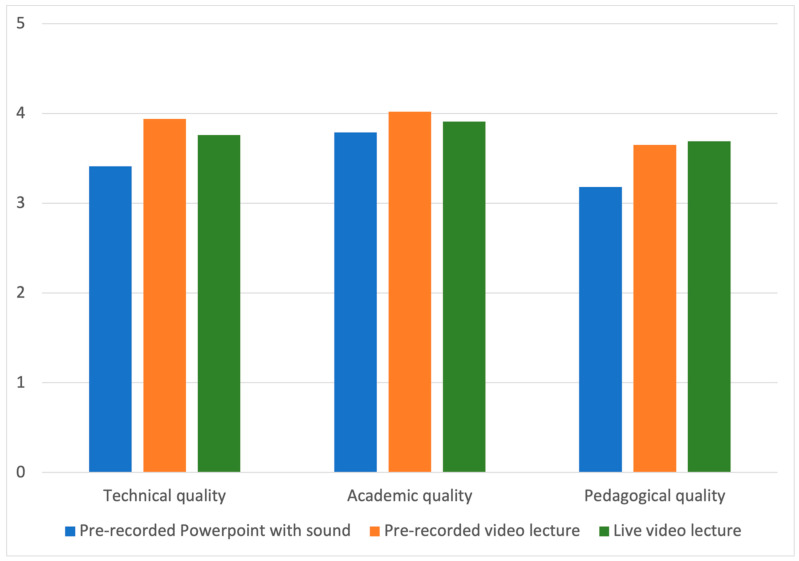
Student experience expressed on a Likert scale regarding technical, academic, and pedagogical quality on pre-recorded PowerPoint with sound, pre-recorded video lectures and live video lectures.

**Figure 2 healthcare-10-00170-f002:**
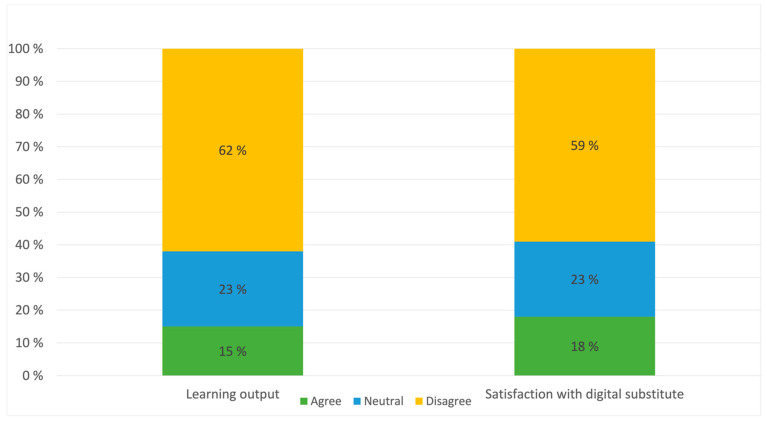
Distribution of agreement, disagreement, or neutral view on statements regarding clinical and practical education.

**Figure 3 healthcare-10-00170-f003:**
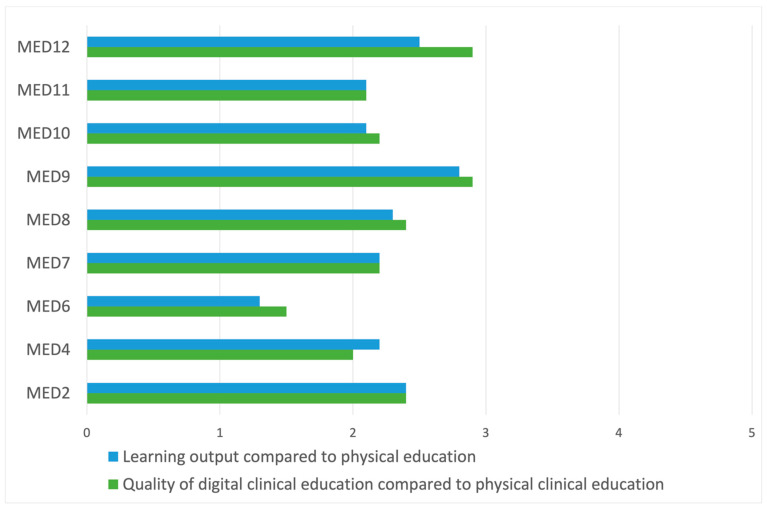
Likert scale mean value on student opinion regarding clinical and practical education depending on semester of study. Learning output: My learning output from digital clinical and practical teaching has been the same as with physical teaching.

**Figure 4 healthcare-10-00170-f004:**
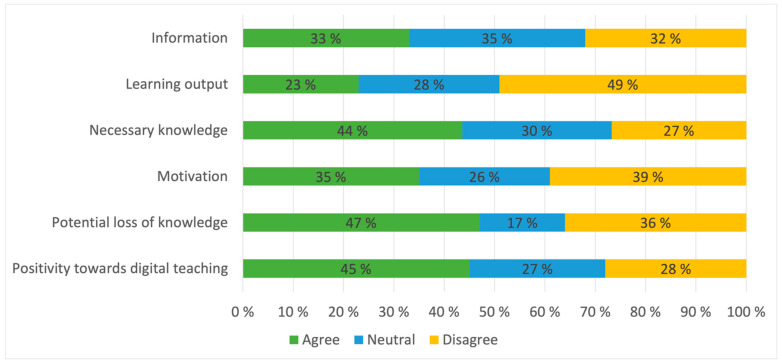
Student attitude to questions regarding their own learning experience this semester. Information: Necessary information about changes in education offer has been given; Learning output: My learning output during digital teaching has been equal to that of physical teaching; Knowledge acquired: I have acquired the necessary amount of knowledge during digital teaching; Motivation: I have been motivated for learning during digital teaching; Potential loss of knowledge: I am anxious that digital teaching has caused me to miss out on important knowledge; Positivity towards digital teaching: I have a positive attitude towards the possibility of digital teaching the next semester.

**Table 1 healthcare-10-00170-t001:** The questionnaire was divided in five main categories. The table illustrates the different categories and includes an example of questions within each category.

Categories	Examples
General situation	How has your living situation been during the COVID-19 pandemic?
Teaching	On a scale of 1–5: How would you rate the quality of PowerPoint with sound?
Your own learning experience	My experience of learning output has been the same as during a normal semester
Exam	I am satisfied with my own achievement on my exam this semester
Digital education as a whole	What I enjoyed most about digital education was

**Table 2 healthcare-10-00170-t002:** An overview of year progression for medical students at UiB. The table includes response rate for the specific year and distribution of each year in the study population.

Year	Semester ^a^	Distribution in Study Population	Response Rate for Each Term
6th year	MED12	39 (17%)	49%
	MED11	9 (8%)	13%
5th year	MED10	11 (5%)	14%
	MED9	13 (6%)	18%
4th year	MED8	36 (16%)	44%
	MED7	17 (7%)	20%
3rd year	MED6	25 (11%)	15%
2nd year	MED4	24 (10%)	15%
1st year	MED2	46 (20%)	27%

^a^ The medical students are in one group of 180 students during the first three years after which they split in two groups of 90 in each.

**Table 3 healthcare-10-00170-t003:** Five most common answers when students were asked about the pros and cons of digital education.

Pros	Cons
Increased opportunity to structure your own studies (67%)	Reduced possibility for socializing with fellow students (35%)
Increased time efficiency (15%)	Less day-to-day structure (17%)
Being able to decide time and place for study (9%)	Lessons deviating from the original timetable and/or being cancelled (14%)
Being more comfortable asking questions during lessons (8%)	Insufficient technical abilities in educators (11%)
Increased availability of lessons (2%)	A general feeling of distance to educator (10%)

**Table 4 healthcare-10-00170-t004:** Five most common answers when students were asked about how to improve digital education in the future.

	Proposals on How to Improve Digital Education	Number of Students
What should the teacher do to improve digital education?	Take advantage of opportunities for interactive teaching	44 (22.9%)
	Familiarize yourself with technical aids before the lesson	39 (20.3%)
	Ensure that students still get their 15-min breaks between lessons	24 (12.5%)
	Ensure that students may still ask questions, despite the lecture being a prerecorded video	19 (9.9%)
	Be sure to record and publish live video lectures	15 (7.8%)
What should the university/faculty do to improve digital education?	Ensure proper education in the use of digital media for educators	61 (35.5%)
	Try to prevent large deviations in scheduled education, and ensure that all teaching receive a digital substitute	46 (26.7%)
	Ensure that information reach the students by establishing a common system for information	23 (13.4%)
	Record and publish all live video lectures	23 (13.4%)
	Optimize video and sound quality	19 (11.0%)

## Data Availability

The data presented in this study are available on request from the corresponding author.

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
