# Peer review of "An Abrupt Transition to Digital Teaching—Norwegian Medical Students and Their Experiences of Learning Output during the Initial Phase of the COVID-19 Lockdown"

_healthcare, 2022, doi:10.3390/healthcare10010170_

Round 1
Reviewer 1 Report
- The authors need to put the number of the participants in the abstract.
- How did the authors distribute the questionnaire? Which sampling technique?
- The proposed conceptual framework lacks in-depth discussion. How did the authors come up with the constructs? What makes their model different from similar studies and how did they ensure the validity of the content?
- Similarly, the research gap of this study is lacking in most updated studies. I cannot justify in what way(s) the past researches failed to provide valuable information regarding this matter.
- Since the authors submitted this study in the Healthcare, I think the manuscript should cite and compare this study in the Introduction and Discussion parts (It is better to do so).
- Prasetyo, Y.T.; Roque, R.A.C.; Chuenyindee, T.; Young, M.N.; Diaz, J.F.T.; Persada, S.F.; Miraja, B.A.; Perwira Redi, A.A.N. Determining Factors Affecting the Acceptance of Medical Education eLearning Platforms during the COVID-19 Pandemic in the Philippines: UTAUT2 Approach. Healthcare2021, 9, 780. https://doi.org/10.3390/healthcare9070780
- If the data collection was conducted more than 1 year, what is the implications of this study?
- How did the authors calculate the reliability and the validity of the questionnaire?
- Which language was used in the questionnaire? Did the authors use two different versions of the questionnaire? If so, how did the authors ensure both versions are compatible with each other?
- Did the authors use any pilot test before conducting the original survey? Using a pilot test is an important step.
- How do the authors justify the sample size? Is there any particular criterion you followed to select the sample size.
Author Response
- The authors need to put the number of the participants in the abstract.
The number of students is now described in the abstract.
- How did the authors distribute the questionnaire? Which sampling technique?
Distribution and sampling of data is now more thoroughly presented in our manuscript.
- The proposed conceptual framework lacks in-depth discussion. How did the authors come up with the constructs? What makes their model different from similar studies and how did they ensure the validity of the content?
The study was conducted at the start of the Covid-19 pandemic to find out how students reacted to the new form of teaching that was introduced, and which both students and teachers had little experience with at that time. The aim of the study was therefor to find out how students had responded and how they reacted both academically and socially to the transition to digital education. The study was therefore constructed as a retrospective study at the end of the semester to evaluate students’ reflections and experiences. Other studies have also looked at the transition to digital teaching, and similarities and differences with other studies are now more extensively discussed in our revised manuscript. The validation of the results is hence based on comparison with other studies.
- Similarly, the research gap of this study is lacking in most updated studies. I cannot justify in what way(s) the past researches failed to provide valuable information regarding this matter.
We believe that context matters, even if similar studies have been conducted elsewhere, it is still of interest how the Norwegian students perceived the situations. The introduction and discussion in our manuscript is now updated with several more recent studies on the subject.
- Since the authors submitted this study in the Healthcare, I think the manuscript should cite and compare this study in the Introduction and Discussion parts (It is better to do so).
- Prasetyo, Y.T.; Roque, R.A.C.; Chuenyindee, T.; Young, M.N.; Diaz, J.F.T.; Persada, S.F.; Miraja, B.A.; Perwira Redi, A.A.N. Determining Factors Affecting the Acceptance of Medical Education eLearning Platforms during the COVID-19 Pandemic in the Philippines: UTAUT2 Approach. Healthcare2021, 9, 780. https://doi.org/10.3390/healthcare9070780
This study is now quoted in our study.
- If the data collection was conducted more than 1 year, what is the implications of this study?
Even if it has taken some time to process the information, we still believe the information can contribute in the shaping of medical education in the future. Data was collected according to the 2020 spring semester, when restrictions were most intense. Medical education in Norway after the spring semester of 2020 have varied between physical education and digital education, depending on year of study and the overall burden of the pandemic in the Norwegian society.
- How did the authors calculate the reliability and the validity of the questionnaire?
We used a self-designed questionnaire Design and layout was discussed several times between the authors. We also used a small test group of five students to maximize face validate.
- Which language was used in the questionnaire? Did the authors use two different versions of the questionnaire? If so, how did the authors ensure both versions are compatible with each other?
Only one version of the questionnaire was used. It was written in Norwegian, and the medical students answered the questionnaire in Norwegian. Norwegian is the first language of Norway and is the first language used in medical education. Questions, tables, and phrases is translated to English for the purpose of this manuscript.
- Did the authors use any pilot test before conducting the original survey? Using a pilot test is an important step.
A small pre-test group of five students were used to standardize questions.
- How do the authors justify the sample size? Is there any particular criterion you followed to select the sample size.
All medical students at UiB (958 in total) were invited to participate, which means there was no sampling.
Reviewer 2 Report
Very relevant and current topic well aligned with this journal
Some Formatting issues, eg: Abstract has 2 different letter types – correct
Introduction can be improved including a stronger discussion of “emergency remote learning” challenges during the pandemic, and implications for students’ learning – see for example:
Khlaif, Z.N., Salha, S. & Kouraichi, B. (2021) Emergency remote learning during COVID-19 crisis: Students’ engagement. Education and Information Technologies 26, 7033–7055 (2021). https://doi.org/10.1007/s10639-021-10566-4
The methods section should also be improved, in particular detailing the process of development and validation of the questionnaire being used, reporting on its validity, reliability and consistency.
Include data reporting/discussing the representativeness of the selected sample – your sample is actually much more representative for the studied population than you seem to realise.
Discuss possible implications for curriculum design in this new online model in order to better support student learning.
Author Response
Very relevant and current topic well aligned with this journal
Some Formatting issues, eg: Abstract has 2 different letter types – correct
- Formatting issues has been resolved
Introduction can be improved including a stronger discussion of “emergency remote learning” challenges during the pandemic, and implications for students’ learning – see for example:
Khlaif, Z.N., Salha, S. & Kouraichi, B. (2021) Emergency remote learning during COVID-19 crisis: Students’ engagement. Education and Information Technologies 26, 7033–7055 (2021). https://doi.org/10.1007/s10639-021-10566-4
- This study has been included in our manuscript, and we have strengthened our introduction with a discussion of emergency remote learning as suggested.
The methods section should also be improved, in particular detailing the process of development and validation of the questionnaire being used, reporting on its validity, reliability and consistency.
- The methods section has been improved and expanded with focus on the development and validity of our study.
Include data reporting/discussing the representativeness of the selected sample – your sample is actually much more representative for the studied population than you seem to realise.
The discussion of representativeness of our selected sample has been modified.
Discuss possible implications for curriculum design in this new online model in order to better support student learning.
The conclusion of our study now includes important focus points to improve digital medical education in the future.
Reviewer 3 Report
The study presents the way in which teaching has had to adapt after the decree of confinement, a topic that can always provide new insights into a phenomenon of interest to the whole educational community.
However, the theoretical justification is scarce and does not allow for an effective contextualisation of how this phenomenon has been dealt with in different contexts.
In addition, instruments that have not been validated are used, making the methodological foundation weak.
These fundamental issues mean that the manuscript cannot be assessed positively.
Author Response
The study presents the way in which teaching has had to adapt after the decree of confinement, a topic that can always provide new insights into a phenomenon of interest to the whole educational community.
However, the theoretical justification is scarce and does not allow for an effective contextualisation of how this phenomenon has been dealt with in different contexts.
In addition, instruments that have not been validated are used, making the methodological foundation weak.
These fundamental issues mean that the manuscript cannot be assessed positively.
Regarding the criticisms from reviewer 3, we are agreeing that the study has some limitations, as also discussed. We have made efforts to strengthen the theoretical justifications of the study in the revised version, as well as added details to provide a more transparent methods section. We believe the aims and principles of the study and hence the results are of interest for the medical education community, as medical education suddenly had to be revised during the covid-19 pandemic. These adaptations are highly contextual which is why we feel snapshots from various settings can provide insight that are transferrable, yet not generalizable, beyond its context of origin. Therefore, we believe our findings to be valid and of interest, and hence our results in the paper to be relevant.
Round 2
Reviewer 1 Report
The authors have successfully addressed my comments. Overall, I think it's a good paper. Congratulations.Author Response
We are grateful for reviewer 1 encoring comments regarding our manuscript.
Reviewer 2 Report
the authors successfully address all comments/suggestions and the manuscript is worthy of being published.
please correct citation format in the new added sections including numbers in () for the new references
Author Response
Thanks for positive comments. The citation and reference list should accordingly be after Healthcare standard mal.
Reviewer 3 Report
Improvements have been detected in both the theoretical background and the methodological description of the study.
Although there is still room for improvement in both aspects, the article is suitable for publication in its current version, considering citation requirements.
Author Response
We are grateful for the comments, and agree that these have improved our manuscript. The citation and reference list should accordingly be after Healthcare standard mal.